# Duality-Induced Regularizer for Tensor Factorization Based Knowledge Graph Completion

**Zhanqiu Zhang**    **Jianyu Cai**    **Jie Wang** *
University of Science and Technology of China
{zzq96,jycai}@mail.ustc.edu.cn,jiewangx@ustc.edu.cn

## Abstract

Tensor factorization based models have shown great power in knowledge graph completion (KGC). However, their performance usually suffers from the overfitting problem seriously. This motivates various regularizers—such as the squared Frobenius norm and tensor nuclear norm regularizers—while the limited applicability significantly limits their practical usage. To address this challenge, we propose a novel regularizer—namely, **DU**ality-induced **R**egul**A**rizer (DURA)—which is not only effective in improving the performance of existing models but widely applicable to various methods. The major novelty of DURA is based on the observation that, for an existing tensor factorization based KGC model (*primal*), there is often another distance based KGC model (*dual*) closely associated with it. Experiments show that DURA yields consistent and significant improvements on benchmarks.

## 1 Introduction

Knowledge graphs contain quantities of factual triplets, which represent structured human knowledge. In the past few years, knowledge graphs have made great achievements in many areas, such as natural language processing [37], question answering [13], recommendation systems [30], and computer vision [18]. Although commonly used knowledge graphs usually contain billions of triplets, they still suffer from the incompleteness problem that a lot of factual triplets are missing. Due to the large scale of knowledge graphs, it is impractical to find all valid triplets manually. Therefore, knowledge graph completion (KGC)—which aims to predict missing links between entities based on known links automatically—has attracted much attention recently.

Distance based (DB) models and tensor factorization based (TFB) models are two important categories of KGC models. DB models use the Minkowski distance to measure the plausibility of a triplet. Although they can achieve state-of-the-art performance, many of them still have difficulty in modeling complex relation patterns, such as one-to-many and many-to-one relations [16, 33]. TFB models treat knowledge graphs as partially observed third-order binary tensors and formulate KGC as a tensor completion problem. Theoretically, these models are highly expressive and can well handle complex relations. However, their performance usually suffers from the overfitting problem seriously and consequently cannot achieve state-of-the-art.

To tackle the overfitting problem of TFB models, researchers propose various regularizers. The squared Frobenius norm regularizer is a popular one that applies to various models [22, 34, 28]. However, experiments show that it may decrease performance for some models (e.g., RESCAL) [23]. More recently, motivated by the great success of the matrix trace norm in the matrix completion problem [25, 5], Lacroix et al. [15] propose a tensor nuclear $p$-norm regularizer. It gains significant improvements against the squared Frobenius norm regularizer. However, it is only suitable for canonical polyadic (CP) decomposition [12] based models, such as CP and ComplEx [28], but

---

not appropriate for a more general class of models, such as RESCAL [22]. Therefore, it is still challenging to find a regularizer that is both widely applicable and effective.

In this paper, we propose a novel regularizer for tensor factorization based KGC models—namely, **DU**ality-induced **R**egul**A**rizer (DURA). The major novelty of DURA is based on the observation called *duality*—for an existing tensor factorization based KGC model (*primal*), there is often another distance based KGC model closely associated with it (*dual*). The duality can be derived by expanding the squared score functions of the associated distance based models. Then, the cross-term in the expansion is exactly a tensor factorization based KGC model, and the squared terms in it give us a regularizer. Using DURA, we can preserve the expressiveness of tensor factorization based KGC models and prevent them from the overfitting problem. DURA is widely applicable to various tensor factorization based models, including CP, ComplEx, and RESCAL. Experiments show that, DURA yields consistent and significant improvements on datasets for the knowledge graph completion task. It is worth noting that, when incorporated with DURA, RESCAL [22]—which is one of the first knowledge graph completion models—performs comparably to state-of-the-art methods and even beats them on several benchmarks.

## 2 Preliminaries

In this section, we review the background of this paper in Section 2.1 and introduce the notations used throughout this paper in Section 2.2.

### 2.1 Background

**Knowledge Graph**  Given a set $\mathcal{E}$ of entities and a set $\mathcal{R}$ of relations, a knowledge graph $\mathcal{K} = \{(e_i, r_j, e_k)\} \subset \mathcal{E} \times \mathcal{R} \times \mathcal{E}$ is a set of triplets, where $e_i$ and $r_j$ are the $i$-th entity and $j$-th relation, respectively. Usually, $e_i$ and $e_k$ are also called the head entity and the tail entity, respectively.

**Knowledge Graph Completion (KGC)**  The goal of KGC is to predict valid but unobserved triplets based on the known triplets in $\mathcal{K}$. KGC models contain two important categories: distance based models and tensor factorization based models, both of which are knowledge graph embedding (KGE) methods. KGE models associate each entity $e_i \in \mathcal{E}$ and relation $r_j \in \mathcal{R}$ with an embedding (may be real or complex vectors, matrices, and tensors) $\mathbf{e}_i$ and $\mathbf{r}_j$. Generally, they define a score function $s : \mathcal{E} \times \mathcal{R} \times \mathcal{E} \to \mathbb{R}$ to associate a score $s(e_i, r_j, e_k)$ with each potential triplet $(e_i, r_j, e_k) \in \mathcal{E} \times \mathcal{R} \times \mathcal{E}$. The scores measure the plausibility of triplets. For a query $(e_i, r_j, ?)$, KGE models first fill the blank with each entity in the knowledge graphs and then score the resulted triplets. Valid triplets are expected to have higher scores than invalid triplets.

**Distance Based (DB) KGC Models**  DB models define the score function $s$ with the Minkowski distance. That is, the score functions have the formulation of $s(e_i, r_j, e_k) = -\|\Gamma(e_i, r_j, e_k)\|_{.p}$, where $\Gamma$ is a model-specific function. Equivalently, we can also use a squared score function $s(e_i, r_j, e_k) = -\|\Gamma(e_i, r_j, e_k)\|_p^2$.

**Tensor Factorization Based (TFB) KGC Models**  TFB models regard a knowledge graph as a third-order binary tensor $\mathcal{X} \in \{0,1\}^{|\mathcal{E}| \times |\mathcal{R}| \times |\mathcal{E}|}$. The $(i, j, k)$ entry $\mathcal{X}_{ijk} = 1$ if $(e_i, r_j, e_k)$ is valid otherwise $\mathcal{X}_{ijk} = 0$. Suppose that $\mathcal{X}_j$ denotes the $j$-th frontal slice of $\mathcal{X}$, i.e., the adjacency matrix of the $j$-th relation. Usually, a TFB KGC model factorizes $\mathcal{X}_j$ as $\mathcal{X}_j \approx \mathbf{Re}\left(\overline{\mathbf{H}}\mathbf{R}_j\mathbf{T}^\top\right)$, where the $i$-th ($k$-th) row of $\mathbf{H}$ ($\mathbf{T}$) is $\mathbf{e}_i$ ($\mathbf{e}_k$), $\mathbf{R}_j$ is a matrix representing relation $r_j$, $\mathbf{Re}\left(\cdot\right)$ and $\bar{\cdot}$ are the real part and the conjugate of a complex matrix, respectively. That is, the score functions are defined as $s(e_i, r_j, e_k) = \mathbf{Re}\left(\bar{\mathbf{e}}_i\mathbf{R}_j\mathbf{e}_k^\top\right)$. Note that the real part and the conjugate of a real matrix are itself. Then, the aim of TFB models is to seek matrices $\mathbf{H}, \mathbf{R}_1, \ldots, \mathbf{R}_{|\mathcal{R}|}, \mathbf{T}$, such that $\mathbf{Re}\left(\overline{\mathbf{H}}\mathbf{R}_j\mathbf{T}^\top\right)$ can approximate $\mathcal{X}_j$. Let $\hat{\mathcal{X}}_j = \mathbf{Re}\left(\overline{\mathbf{H}}\mathbf{R}_j\mathbf{T}^\top\right)$ and $\hat{\mathcal{X}}$ be a tensor of which the $j$-th frontal slice is $\hat{\mathcal{X}}_j$. The regularized formulation of a tensor factorization based model can be written as

$$\min_{\hat{\mathcal{X}}_1, \ldots, \hat{\mathcal{X}}_{|\mathcal{R}|}} \sum_{j=1}^{|\mathcal{R}|} L(\mathcal{X}_j, \hat{\mathcal{X}}_j) + \lambda g(\hat{\mathcal{X}}), \tag{1}$$

where $\lambda > 0$ is a fixed parameter, $L(\mathcal{X}_j, \hat{\mathcal{X}}_j)$ measures the discrepancy between $\mathcal{X}_j$ and $\hat{\mathcal{X}}_j$, and $g$ is the regularization function.

## 2.2 Other Notations

We use $h_i \in \mathcal{E}$ and $t_k \in \mathcal{E}$ to distinguish head and tail entities. Let $\|\cdot\|_1$, $\|\cdot\|_2$, and $\|\cdot\|_F$ denote the $L_1$ norm, the $L_2$ norm, and the Frobenius norm of matrices or vectors. We use $\langle\cdot,\cdot\rangle$ to represent the inner products of two real or complex vectors. Specifically, if $\mathbf{u}, \mathbf{v} \in \mathbb{C}^{1\times n}$ are two row vectors in the complex space, then the inner product is defined as $\langle\mathbf{u},\mathbf{v}\rangle = \bar{\mathbf{u}}\mathbf{v}^\top$.

# 3 Related Work

Knowledge graph completion (KGC) models include rule-based methods [9, 35], KGE methods, and hybrid methods [10]. This work is related to KGE methods [4, 28, 21, 36]. More specifically, it is related to distance based KGE models and tensor factorization based KGE models.

Distance based models describe relations as relational maps between head and tail entities. Then, they use the Minkowski distance to measure the plausibility of a given triplet. For example, TransE [4] and its variants [33, 16] represent relations as translations in vector spaces. They assume that a valid triplet $(h_i, r_j, t_k)$ satisfies $\mathbf{h}_{i,r_j} + \mathbf{r}_j \approx \mathbf{t}_{k,r_j}$, where $\mathbf{h}_{i,r_j}$ and $\mathbf{t}_{k,r_j}$ mean that entity embeddings may be relation-specific. Structured embedding (SE) [3] uses linear maps to represent relations. Its score function is defined as $s(h_i, r_j, t_k) = -\|\mathbf{R}_j^1\mathbf{h}_i - \mathbf{R}_j^2\mathbf{t}_k\|_1$. RotatE [26] defines each relation as a rotation in a complex vector space and the score function is defined as $s(h_i, r_j, t_k) = -\|\mathbf{h}_i \circ \mathbf{r}_j - \mathbf{t}_k\|_1$, where $\mathbf{h}_i, \mathbf{r}_j, \mathbf{t}_k \in \mathbb{C}^k$ and $|[\mathbf{r}]_i| = 1$. ModE [38] assumes that $\mathbf{R}_j^1$ is diagonal and $\mathbf{R}_j^2$ is an identity matrix. It shares a similar score function $s(h_i, r_j, t_k) = -\|\mathbf{h}_i \circ \mathbf{r}_j - \mathbf{t}_k\|_1$ with RotatE but $\mathbf{h}_i, \mathbf{r}_j, \mathbf{t}_k \in \mathbb{R}^k$.

Tensor factorization based models formulate the KGC task as a third-order binary tensor completion problem. RESCAL [22] factorizes the $j$-th frontal slice of $\mathcal{X}$ as $\mathcal{X}_j \approx \mathbf{A}\mathbf{R}_j\mathbf{A}^\top$, in which embeddings of head and tail entities are from the same space. As the relation specific matrices contain lots of parameters, RESCAL is prone to be overfitting. DistMult [34] simplifies the matrix $\mathbf{R}_j$ in RESCAL to be diagonal, while it sacrifices the expressiveness of models and can only handle symmetric relations. In order to model asymmetric relations, ComplEx [28] extends DistMult to complex embeddings. Both DistMult and ComplEx can be regarded as variants of CP decomposition [12], which are in real and complex vector spaces, respectively.

Tensor factorization based (TFB) KGC models usually suffer from overfitting problem seriously, which motivates various regularizers. In the original papers of TFB models, the authors usually use the squared Frobenius norm ($L_2$ norm) regularizer [22, 34, 28]. This regularizer cannot bring satisfying improvements. Consequently, TFB models do not gain comparable performance to distance based models [26, 38]. More recently, Lacroix et al. [15] propose to use the tensor nuclear 3-norm [8] (N3) as a regularizer, which brings more significant improvements than the squared Frobenius norm regularizer. However, it is designed for the CP-like models, such as CP and ComplEx, and not suitable for more general models such as RESCAL. Moreover, some regularization methods aim to leverage external background knowledge [19, 7, 20]. For example, to model equivalence and inversion axioms, Minervini et al. [19] impose a set of model-dependent soft constraints on the predicate embeddings. Ding et al. [7] use non-negativity constraints on entity embeddings and approximate entailment constraints on relation embeddings to impose prior beliefs upon the structure of the embeddings space.

# 4 Methods

In this section, we introduce a novel regularizer—**DU**ality-induced **R**egul**A**rizer (**DURA**)—for tensor factorization based knowledge graph completion. We first introduce basic DURA in Section 4.1 and explain why it is effective in Section 4.2. Then, we introduce DURA in Section 4.3. Finally, we give a theoretical analysis for DURA under some special cases in Section 4.4.

## 4.1 Basic DURA

Consider the knowledge graph completion problem $(h_i, r_j, ?)$. That is, we are given the head entity and the relation, aiming to predict the tail entity. Suppose that $f_j(i, k)$ measures the plausibility of a

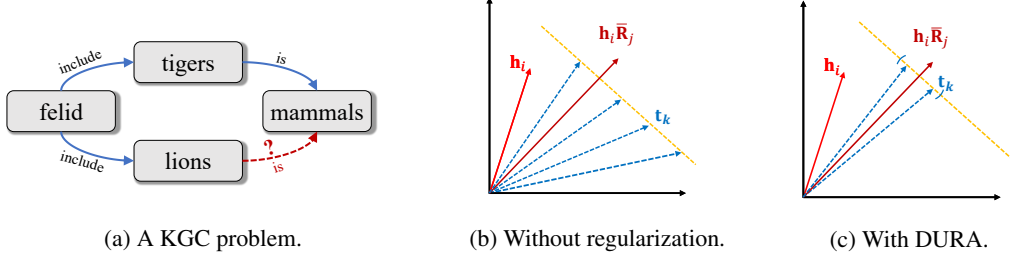

| (a) A KGC problem. | (b) Without regularization. | (c) With DURA. |

Figure 1: An illustration of why basic DURA can improve the performance of TFB models when the embedding dimensions are 2. Suppose that triplets $(h_i, r_j, t_k)$ $(k = 1, 2, \ldots, n)$ are valid. (a) Figure 1a demonstrates that tail entities connected to a head entity through the same relation should have similar embeddings. (b) Figure 1b shows that TFB models without regularization can get the same score even though the embeddings of $t_k$ are dissimilar. (c) Figure 1c shows that with DURA, embeddings of $t_k$ are encouraged to locate in a small region.

given triplet $(h_i, r_j, t_k)$, i.e., $f_j(i, k) = s(h_i, r_j, t_k)$. Then the score function of a TFB model is

$$f_j(i, k) = \mathbf{Re}\left(\overline{\mathbf{h}}_i \mathbf{R}_j \mathbf{t}_k^\top\right) = \mathbf{Re}\left(\langle \mathbf{h}_i \overline{\mathbf{R}}_j, \mathbf{t}_k \rangle\right). \tag{2}$$

It first maps the entity embeddings $\mathbf{h}_i$ by a linear transformation $\overline{\mathbf{R}}_j$, and then uses the real part of an inner product to measure the similarity between $\mathbf{h}_i \overline{\mathbf{R}}_j$ and $\mathbf{t}_k$. Notice that another commonly used similarity measure—the squared Euclidean distance—can replace the inner product similarity in Equation (2). We can obtain an associated distance based model formulated as

$$f_j^E(i, k) = -\|\mathbf{h}_i \overline{\mathbf{R}}_j - \mathbf{t}_k\|_2^2. \tag{3}$$

Therefore, there exists a ***duality***: for an existing tensor factorization based KGC model (***primal***), there is often another distance based KGC model (***dual***) closely associated with it.

Specifically, the relationship between the primal and the dual can be formulated as

$$\begin{aligned} f_j^E(i, k) &= -\|\mathbf{h}_i \overline{\mathbf{R}}_j - \mathbf{t}_k\|_2^2 \\ &= -\|\mathbf{h}_i \overline{\mathbf{R}}_j\|_2^2 - \|\mathbf{t}_k\|_2^2 + 2\mathbf{Re}\left(\langle \mathbf{h}_i \overline{\mathbf{R}}_j, \mathbf{t}_k \rangle\right) \\ &= 2f_j(i, k) - \|\mathbf{h}_i \overline{\mathbf{R}}_j\|_2^2 - \|\mathbf{t}_k\|_2^2. \end{aligned} \tag{4}$$

Usually, we expect $f_j^E(i, k)$ and $f_j(i, k)$ to be higher for all valid triplets $(h_i, r_j, t_k)$ than those for invalid triplets. Suppose that $\mathcal{S}$ is the set that contains all valid triplets. Then, for triplets in $\mathcal{S}$, we have that

$$\begin{aligned} \max f_j^E(i, k) &= \min -f_j^E(i, k) \\ &= \min -2f_j(i, k) + \|\mathbf{h}_i \overline{\mathbf{R}}_j\|_2^2 + \|\mathbf{t}_k\|_2^2. \end{aligned} \tag{5}$$

By noticing that $\min -2f_j(i, k) = \max 2f_j(i, k)$ is exactly the aim of a TFB model, the duality induces a regularizer for tensor factorization based KGC models, i.e.,

$$\sum_{(h_i, r_j, t_k) \in \mathcal{S}} \|\mathbf{h}_i \overline{\mathbf{R}}_j\|_2^2 + \|\mathbf{t}_k\|_2^2, \tag{6}$$

which is called basic DURA.

## 4.2 Why Basic DURA Helps

In this section, we demonstrate that basic DURA encourages tail entities connected to a head entity through the same relation to have similar embeddings, which accounts for its effectiveness in improving performance of TFB models.

First, we claim that tail entities connected to a head entity through the same relation should have similar embeddings. Suppose that we know a head entity $h_i$ and a relation $r_j$, and our aim is to predict the tail entity. If $r_j$ is a one-to-many relation, i.e., there exist two entities $t_1$ and $t_2$ such

that both $(h_i, r_j, t_1)$ and $(h_i, r_j, t_2)$ are valid, then we expect that $t_1$ and $t_2$ have similar semantics. For example, if two triplets (`felid`, `include`, `tigers`) and (`felid`, `include`, `lions`) are valid, then `tigers` and `lions` should have similar semantics. Further, we expect that entities with similar semantics have similar embeddings. In this way, if we have known that (`tigers`, `is`, `mammals`) is valid, then we can predict that (`lions`, `is`, `mammals`) is also valid. See Figure 1a for an illustration of the prediction process.

However, TFB models fail to achieve the above goal. As shown in Figure 1b, suppose that we have known $\mathbf{h}_i \bar{\mathbf{R}}_j$ when the embedding dimension is 2. Then, we can get the same score $s(h_i, r_j, t_k)$ for $k = 1, 2, \ldots, n$ so long as $\mathbf{t}_k$ lies on the same line perpendicular to $\mathbf{h}_i \bar{\mathbf{R}}_j$. Generally, the entities $t_1$ and $t_2$ have similar semantics. However, their embeddings $\mathbf{t}_1$ and $\mathbf{t}_2$ can even be orthogonal, which means that the two embeddings are dissimilar. Therefore, the performance of TFB models for knowledge graph completion is usually unsatisfying.

By Equation (5), we know that basic DURA constraints the distance between $\mathbf{h}_i \bar{\mathbf{R}}_j$ and $\mathbf{t}_k$. When $\mathbf{h}_i$ and $\bar{\mathbf{R}}_j$ are known, $\mathbf{t}_k$ lies in a small region (see Figure 1c and we verify this claim in Section 5.4). Therefore, tail entities connected to a head entity through the same relation will have similar embeddings, which is beneficial to the prediction of unknown triplets.

### 4.3 DURA

Basic DURA encourages tail entities with similar semantics to have similar embeddings. However, it cannot handle the case that head entities have similar semantics.

Suppose that two triplets (`tigers`, `is`, `mammals`) and (`lions`, `is`, `mammals`) are valid. Similar to the discussion in Section 4.2, we expect that `tigers` and `lions` have similar semantics and thus have similar embeddings. If we further know that (`feild`, `include`, `tigers`) is valid, we can predict that (`feild`, `include`, `lions`) is valid. However, basic DURA cannot handle the case. Let $\mathbf{h}_1, \mathbf{h}_2, \mathbf{t}_1$, and $\mathbf{R}_1$ be the embeddings of `tigers`, `lions`, `mammals`, and `is`, respectively. Then, $\mathbf{Re}\,(\bar{\mathbf{h}}_1 \mathbf{R}_1 \mathbf{t}_1^\top)$ and $\mathbf{Re}\,(\bar{\mathbf{h}}_2 \mathbf{R}_1 \mathbf{t}_1^\top)$ can be equal even if $\mathbf{h}_1$ and $\mathbf{h}_2$ are orthogonal, as long as $\mathbf{h}_1 \bar{\mathbf{R}}_1 = \mathbf{h}_2 \bar{\mathbf{R}}_1$.

To tackle the above issue, noticing that $\mathbf{Re}\,(\bar{\mathbf{h}}_i \mathbf{R}_j \mathbf{t}_k^\top) = \mathbf{Re}\,(\bar{\mathbf{t}}_k \bar{\mathbf{R}}_j^\top \mathbf{h}_i^\top)$, we define another dual distance based KGC model

$$\tilde{f}_j^E(i, k) = -\|\mathbf{t}_k \mathbf{R}_j^\top - \mathbf{h}_i\|_2^2,$$

Then, similar to the derivation in Equation (5), the duality induces a regularizer given by

$$\sum_{(h_i, r_j, t_k) \in \mathcal{S}} \|\mathbf{t}_k \mathbf{R}_j^\top\|^2 + \|\mathbf{h}_i\|^2. \tag{7}$$

When a TFB model are incorporated with regularizer (7), head entities with similar semantics will have similar embeddings.

Finally, combining the regularizer (6) and (7), DURA has the form of

$$\sum_{(h_i, r_j, t_k) \in \mathcal{S}} \left[ \|\mathbf{h}_i \bar{\mathbf{R}}_j\|_2^2 + \|\mathbf{t}_k\|_2^2 + \|\mathbf{t}_k \mathbf{R}_j^\top\|_2^2 + \|\mathbf{h}_i\|_2^2 \right]. \tag{8}$$

### 4.4 Theoretic Analysis for Diagonal Relation Matrices

If we further relax the summation condition in the regularizer (8) to all possible entities and relations, we can write DURA as:

$$|\mathcal{E}| \sum_{j=1}^{|\mathcal{R}|} (\|\mathbf{H}\bar{\mathbf{R}}_j\|_F^2 + \|\mathbf{T}\|_F^2 + \|\mathbf{T}\mathbf{R}_j^\top\|_F^2 + \|\mathbf{H}\|_F^2), \tag{9}$$

where $|\mathcal{E}|$ and $|\mathcal{R}|$ are the number of entities and relations, respectively.

In the rest of this section, we use the same definitions of $\hat{\mathcal{X}}_j$ and $\hat{\mathcal{X}}$ as in the problem (1). When the relation embedding matrices $\mathbf{R}_j$ are diagonal in $\mathbb{R}$ or $\mathbb{C}$ as in CP or ComplEx, the formulation (9) gives an upper bound to the tensor nuclear 2-norm of $\hat{\mathcal{X}}$, which is an extension of trace norm regularizers in matrix completion. To simplify the notations, we take CP as an example, in which all involved embeddings are real. The conclusion in complex space can be analogized accordingly.

**Definition 1** (Friedland & Lim [8]). *The nuclear 2-norm of a 3D tensor $\mathcal{A} \in \mathbb{R}^{n_1} \otimes \mathbb{R}^{n_2} \otimes \mathbb{R}^{n_3}$ is*

$$\|\mathcal{A}\|_* = \min \left\{ \sum_{i=1}^{r} \|\boldsymbol{u}_{1,i}\|_2 \|\boldsymbol{u}_{2,i}\|_2 \|\boldsymbol{u}_{3,i}\|_2 : \mathcal{A} = \sum_{i=1}^{r} \boldsymbol{u}_{1,i} \otimes \boldsymbol{u}_{2,i} \otimes \boldsymbol{u}_{3,i}, r \in \mathbb{N} \right\},$$

*where $\boldsymbol{u}_{k,i} \in \mathbb{R}^{n_k}$ for $k = 1,...,3$, $i = 1,...,r$, and $\otimes$ denotes the outer product.*

For notation convenience, we define a relation matrix $\widetilde{\mathbf{R}} \in \mathbb{R}^{|\mathcal{R}| \times D}$, of which the $j$-th row consists of the diagonal entries of $\mathbf{R}_j$. That is, $\widetilde{\mathbf{R}}(j,d) = \mathbf{R}_j(d,d)$, where $\mathbf{R}(i,j)$ represents the entry in the $i$-th row and $j$-th column of the matrix $\mathbf{R}$.

In the knowledge graph completion problem, the tensor nuclear 2-norm of $\hat{\mathcal{X}}$ is

$$\|\hat{\mathcal{X}}\|_* = \min \left\{ \sum_{d=1}^{D} \|\mathbf{h}_{:d}\|_2 \|\mathbf{r}_{:d}\|_2 \|\mathbf{t}_{:d}\|_2 : \hat{\mathcal{X}} = \sum_{d=1}^{D} \mathbf{h}_{:d} \otimes \mathbf{r}_{:d} \otimes \mathbf{t}_{:d} \right\},$$

where $D$ is the embedding dimension, $\mathbf{h}_{:d}$, $\mathbf{r}_{:d}$, and $\mathbf{t}_{:d}$ are the $d$-th columns of $\mathbf{H}$, $\widetilde{\mathbf{R}}$, and $\mathbf{T}$.

For DURA in (9), we have the following theorem.

**Theorem 1.** *Suppose that $\hat{\mathcal{X}}_j = \boldsymbol{H}\boldsymbol{R}_j\boldsymbol{T}^\top$ for $j = 1, 2, \ldots, |\mathcal{R}|$, where $\boldsymbol{H}, \boldsymbol{T}, \boldsymbol{R}_j$ are real matrices and $\boldsymbol{R}_j$ is diagonal. Then, the following equation holds*

$$\min_{\hat{\mathcal{X}}_j = \boldsymbol{H}\boldsymbol{R}_j\boldsymbol{T}^\top} \frac{1}{\sqrt{|\mathcal{R}|}} \sum_{j=1}^{|\mathcal{R}|} (\|\boldsymbol{H}\boldsymbol{R}_j\|_F^2 + \|\boldsymbol{T}\|_F^2 + \|\boldsymbol{T}\boldsymbol{R}_j^\top\|_F^2 + \|\boldsymbol{H}\|_F^2) = \|\hat{\mathcal{X}}\|_*.$$

*The minimization attains if and only if $\|\boldsymbol{h}_{:d}\|_2 \|\boldsymbol{r}_{:d}\|_2 = \sqrt{|\mathcal{R}|}\|\boldsymbol{t}_{:d}\|_2$ and $\|\boldsymbol{t}_{:d}\|_2 \|\boldsymbol{r}_{:d}\|_2 = \sqrt{|\mathcal{R}|}\|\boldsymbol{h}_{:d}\|_2$, $\forall d \in \{1, 2, \ldots, D\}$, where $\boldsymbol{h}_{:d}$, $\boldsymbol{r}_{:d}$, and $\boldsymbol{t}_{:d}$ are the $d$-th columns of $\boldsymbol{H}$, $\widetilde{\boldsymbol{R}}$, and $\boldsymbol{T}$, respectively.*

*Proof.* See the supplementary material. □

Therefore, DURA in (9) gives an upper bound to the tensor nuclear 2-norm, which is a tensor analog to the matrix trace norm.

**Remark** DURA in (8) is actually a weighted version of the one in (9), in which the regularization terms corresponding to the sampled valid triplets. As shown in Srebro & Salakhutdinov [24] and Lacroix et al. [15], the weighted versions of regularizers usually outperform the unweighted regularizer when entries of the matrix or tensor are sampled non-uniformly. Therefore, in the experiments, we implement DURA in a weighted way as in (8).

## 5 Experiments

In this section, we introduce the experimental settings in Section 5.1 and show the effectiveness of DURA in Section 5.2. We compare DURA to other regularizers in Section 5.3 and visualize the entity embeddings in Section 5.4. Finally, we analyze the sparsity induced by DURA in Section 5.5. The code of HAKE is available on GitHub at `https://github.com/MIRALab-USTC/KGE-DURA`.

### 5.1 Experimental Settings

We consider three public knowledge graph datasets—WN18RR [27], FB15k-237 [6], and YAGO3-10 [17] for the knowledge graph completion task, which have been divided into training, validation, and testing set in previous works. The statistics of these datasets are shown in Table 1. WN18RR, FB15k-237, and YAGO3-10 are extracted from WN18 [4], FB15k [4], and YAGO3 [17], respectively. Toutanova & Chen [27] and Dettmers et al. [6] indicated the test set leakage problem in WN18 and FB15k, where some

Table 1: Statistics of three benchmark datasets.

|  | WN18RR | FB15k-237 | YAGO3-10 |
|---|---|---|---|
| #Entity | 40,943 | 14,541 | 123,182 |
| #Relation | 11 | 237 | 37 |
| #Train | 86,835 | 272,115 | 1,079,040 |
| #Valid | 3,034 | 17,535 | 5,000 |
| #Test | 3,134 | 20,466 | 5,000 |

test triplets may appear in the training dataset in the form of reciprocal relations. They created WN18RR and FB15k-237 to avoid the test set leakage problem, and we use them as the benchmark datasets. We use MRR and Hits@N (H@N) as evaluation metrics. For more details of training and evaluation protocols, please refer to the supplementary material.

Moreover, we find it better to assign different weights for the parts involved with relations. That is, the optimization problem has the form of

$$\min \sum_{(e_i, r_j, e_k) \in \mathcal{S}} [\ell_{ijk}(\mathbf{H}, \mathbf{R}_1, \ldots, \mathbf{R}_J, \mathbf{T})$$
$$+ \lambda(\lambda_1(\|\mathbf{h}_i\|_2^2 + \|\mathbf{t}_k\|_2^2) + \lambda_2(\|\mathbf{h}_i \overline{\mathbf{R}}_j\|_2^2 + \|\mathbf{t}_k \mathbf{R}_j^\top\|_2^2))],$$

where $\lambda, \lambda_1, \lambda_2 > 0$ are fixed hyperparameters. We search $\lambda$ in $\{0.005, 0.01, 0.05, 0.1, 0.5\}$ and $\lambda_1, \lambda_2$ in $\{0.5, 1.0, 1.5, 2.0\}$.

Table 2: Evaluation results on WN18RR, FB15k-237 and YAGO3-10 datasets. We reimplement CP, DistMult, ComplEx, and RESCAL using the "reciprocal" setting [15, 14], which leads to better results than the reported results in the original paper.

| | WN18RR | | | FB15k-237 | | | YAGO3-10 | | |
|---|---|---|---|---|---|---|---|---|---|
| | MRR | H@1 | H@10 | MRR | H@1 | H@10 | MRR | H@1 | H@10 |
| RotatE | .476 | .428 | .571 | .338 | .241 | .533 | .495 | .402 | .670 |
| MuRP | .481 | .440 | .566 | .335 | .243 | .518 | - | - | - |
| HAKE | .497 | .452 | **.582** | .346 | .250 | .542 | .546 | .462 | .694 |
| TuckER | .470 | .443 | .526 | .358 | .266 | .544 | - | - | - |
| CP | .438 | .414 | .485 | .333 | .247 | .508 | .567 | .494 | .698 |
| RESCAL | .455 | .419 | .493 | .353 | .264 | .528 | .566 | .490 | .701 |
| ComplEx | .460 | .428 | .522 | .346 | .256 | .525 | .573 | .500 | .703 |
| CP-DURA | .478 | .441 | .552 | .367 | .272 | .555 | .579 | .506 | .709 |
| RESCAL-DURA | **.498** | **.455** | .577 | .368 | **.276** | .550 | .579 | .505 | .712 |
| ComplEx-DURA | .491 | .449 | .571 | **.371** | **.276** | **.560** | **.584** | **.511** | **.713** |

## 5.2 Main Results

In this section, we compare the performance of DURA against several state-of-the-art KGC models, including CP [12], RESCAL [22], ComplEx [28], TuckER [2] and some DB models: RotatE [26], MuRP [1], and HAKE [38].

Table 2 shows the effectiveness of DURA. RESCAL-DURA and ComplEx-DURA perform competitively with the SOTA DB models. RESCAL-DURA outperforms all the compared DB models in terms of MRR and H@1. Note that we reimplement CP, ComplEx, and RESCAL under the "reciprocal" setting [14, 15], and obtain better results than the reported performance in the original papers. Overall, TFB models with DURA significantly outperform those without DURA, which shows its effectiveness in preventing models from overfitting.

Generally, models with more parameters and datasets with smaller sizes imply a larger risk of overfitting. Among the three datasets, WN18RR has the smallest size of only 11 kinds of relations and around $80k$ training samples. Therefore, the improvements brought by DURA on WN18RR are expected to be larger compared with other datasets, which is consistent with the experiments. As stated in Wang et al. [31], RESCAL is a more expressive model, but it is prone to overfit on small- and medium-sized datasets because it represents relations with much more parameters. For example, on WN18RR dataset, RESCAL gets an H@10 score of 0.493, which is lower than ComplEx (0.522). The advantage of its expressiveness does not show up at all. However, incorporated with DURA, RESCAL gets an 8.4% improvement on H@10 and finally attains 0.577, which outperforms all compared models. On larger datasets such as YAGO3-10, overfitting also exists but may be non-significant. Nonetheless, DURA still leads to consistent improvement, demonstrating the ability of DURA to prevent models from overfitting.

Table 3: Comparison between DURA, the squared Frobenius norm (FRO), and the nuclear 3-norm (N3) regularizers. Results of * are taken from Lacroix et al. [15]. CP-N3 and ComplEx-N3 are re-implemented and their performances are better than the reported results in Lacroix et al. [15]. The best performance on each model are marked in bold.

| | WN18RR | | | FB15k-237 | | | YAGO3-10 | | |
|---|---|---|---|---|---|---|---|---|---|
| | MRR | H@1 | H@10 | MRR | H@1 | H@10 | MRR | H@1 | H@10 |
| CP-FRO* | .460 | - | .480 | .340 | - | .510 | .540 | - | .680 |
| CP-N3 | .470 | .430 | .544 | .354 | .261 | .544 | .577 | .505 | .705 |
| CP-DURA | **.478** | **.441** | **.552** | **.367** | **.272** | **.555** | **.579** | **.506** | **.709** |
| ComplEx-FRO* | .470 | - | .540 | .350 | - | .530 | .570 | - | .710 |
| ComplEx-N3 | .489 | .443 | **.580** | .366 | .271 | .558 | .577 | .502 | .711 |
| ComplEx-DURA | **.491** | **.449** | .571 | **.371** | **.276** | **.560** | **.584** | **.511** | **.713** |
| RESCAL-FRO | .397 | .363 | .452 | .323 | .235 | .501 | .474 | .392 | .628 |
| RESCAL-DURA | **.498** | **.455** | **.577** | **.368** | **.276** | **.550** | **.579** | **.505** | **.712** |

## 5.3 Comparison to Other Regularizers

In this section, we compare DURA to the popular squared Frobenius norm regularizer and the recent tensor nuclear 3-norm (N3) regularizer [15]. The squared Frobenius norm regularizer is given by $g(\hat{\mathcal{X}}) = \|\mathbf{H}\|_F^2 + \|\mathbf{T}\|_F^2 + \sum_{j=1}^{|\mathcal{R}|}\|\mathbf{R}_j\|_F^2$. N3 regularizer is given by $g(\hat{\mathcal{X}}) = \sum_{d=1}^{D}(\|\mathbf{h}_{:d}\|_3^3 + \|\mathbf{r}_{:d}\|_3^3 + \|\mathbf{t}_{:d}\|_3^3)$, where $\|\cdot\|_3$ denotes $L_3$ norm of vectors.

We implement both the squared Frobenius norm (FRO) and N3 regularizers in the weighted way as stated in Lacroix et al. [15]. Table 3 shows the performance of the three regularizers on three popular models: CP, ComplEx, and RESCAL. Note that when the TFB model is RESCAL, we only compare DURA to the squared Frobenius norm regularization as N3 does not apply to it.

For CP and ComplEx, DURA brings consistent improvements compared to FRO and N3 on all datasets. Specifically, on FB15k-237, compared to CP-N3, CP-DURA gets an improvement of 0.013 in terms of MRR. Even for the previous state-of-the-art TFB model ComplEx, DURA brings further improvements against the N3 regularizer. Incorporated with FRO, RESCAL performs worse than the vanilla model, which is consistent with the results in Ruffinelli et al. [23]. However, RESCAL-DURA brings significant improvements against RESCAL. All the results demonstrate that DURA is more widely applicable than N3 and more effective than the squared Frobenius norm regularizer.

## 5.4 Visualization

In this section, we visualize the tail entity embeddings using T-SNE [29] to show that DURA encourages tail entities with similar semantics to have similar embeddings.

Suppose that $(h_i, r_j)$ is a *query*, where $h_i$ and $r_j$ are head entities and relations, respectively. An entity $t_k$ is an *answer* to a query $(h_i, r_j)$ if $(h_i, r_j, t_k)$ is valid. We randomly selected 10 queries in FB15k-237, each of which has more than 50 answers. [1] Then, we use T-SNE to visualize the answers' embeddings generated by CP and CP-DURA. Figure 2 shows the visualization results. Each entity is represented by a 2D point and points in the same color

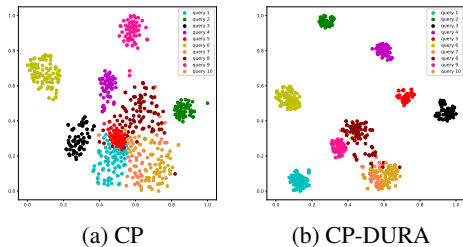

(a) CP　　　　　　(b) CP-DURA

Figure 2: Visualization of the embeddings of tail entities using T-SNE. A point represents a tail entity. Points in the same color represent tail entities that have the same $(h_r, r_j)$ context.

represent tail entities with the same $(h_i, r_j)$ context (i.e. query). Figure 2 shows that, with DURA, entities with the same $(h_i, r_j)$ contexts are indeed being assigned more similar representations, which verifies the claims in Section 4.2.

## 5.5 Sparsity Analysis

As real-world knowledge graphs usually contain billions of entities, the storage of entity embeddings faces severe challenges. Intuitively, if embeddings are sparse, that is, most of the entries are zero, we can store them with less storage. Therefore, the sparsity of the generated entity embeddings becomes crucial for real-world applications. In this part, we analyze the sparsity of embeddings induced by different regularizers.

Generally, there are few entries of entity embeddings that are exactly equal to 0 after training, which means that it is hard to obtain sparse entity embeddings directly. However, when we score triplets using the trained model, the embedding entries with values close to 0 will have minor contributions to the score of a triplet. If we set the embedding entries close to 0 to be exactly 0, we can transform embeddings into sparse ones. Thus, there is a trade-off between sparsity and performance decrement.

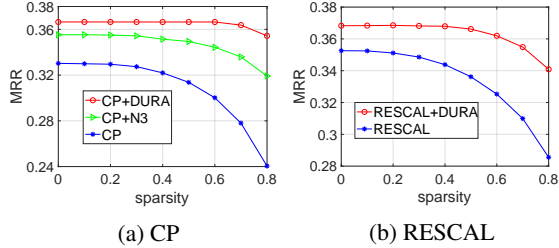

(a) CP  (b) RESCAL

Figure 3: The effect of entity embeddings' $\lambda$-sparsity on MRR. The used dataset is FB15k-237.

We define the following $\lambda$-sparsity to indicate the proportion of entries that are close to zero:

$$s_\lambda = \frac{\sum_{i=1}^{I} \sum_{d=1}^{D} \mathbb{1}_{\{|x|<\lambda\}}(\mathbf{E}_{id})}{I \times D}, \tag{10}$$

where $\mathbf{E} \in \mathbb{R}^{I \times D}$ is the entity embedding matrix, $\mathbf{E}_{id}$ is the entry in the $i$-th row and $d$-th column of $\mathbf{E}$, $I$ is the number of entities, $D$ is the embedding dimension, and $\mathbb{1}_{\mathcal{C}}(x)$ is the indicator function that takes value of 1 if $x \in \mathcal{C}$ or otherwise the value of 0.

To generate sparse version entity embeddings, following Equation (10), we select all the entries of entity embeddings—of which the absolute value are less than a threshold $\lambda$—and set them to be 0. Note that for any given $s_\lambda$, we can always find a proper threshold $\lambda$ to approximate it, as the formula is increasing with respect to $\lambda$. Then, we evaluate the quality of sparse version entity embeddings on the knowledge graph completion task. Figure 3 shows the effect of entity embeddings' $\lambda$-sparsity on MRR. Results in the figure show that DURA causes much gentler performance decrement as the embedding sparsity increases. In Figure 3a, incorporated with DURA, CP maintains MRR of 0.366 unchanged even when 60% entries are set to 0. More surprisingly, when the sparsity reaches 70%, CP-DURA can still outperform CP-N3 with zero sparsity. For RESCAL, when setting 80% entries to be 0, RESCAL-DURA still has the MRR of 0.341, which significantly outperforms vanilla RESCAL, whose MRR has decreased from 0.352 to 0.286. In a word, incorporating with DURA regularizer, the performance of CP and RESCAL remains comparable to the state-of-the-art models, even when 70% of entity embeddings' entries are set to 0.

Following Han et al. [11], we store the sparse version embedding matrices using compressed sparse row (CSR) format or compressed sparse column (CSC) format, which requires $2a + n + 1$ numbers, where $a$ is the number of non-zero elements and $n$ is the number of rows or columns. Experiments show that DURA brings about 65% fewer storage costs for entity embeddings when 70% of the entries are set to 0. Therefore, DURA can significantly reduce the storage usage while maintaining satisfying performance.

## 6 Conclusion

We propose a widely applicable and effective regularizer—namely, DURA—for tensor factorization based knowledge graph completion models. DURA is based on the observation that, for an existing tensor factorization based KGC model (primal), there is often another distance based KGC model (dual) closely associated with it. Experiments show that DURA brings consistent and significant improvements to TFB models on benchmark datasets. Moreover, visualization resultls show that DURA can encourage entities with similar semantics to have similar embeddings, which is beneficial to the prediction of unknown triplets. Since the current formulation of DURA is designed for tensor factorization based models, a direction for future work is to extend DURA to models in other categories, such as KBAT [21] and GAATs [32].

## Broader Impact

The proposed regularizer DURA can significantly improve the performance of tensor factorization based knowledge graph completion models. In other words, it can help us to predict missing links in knowledge graphs automatically. Therefore, using models with DURA, we do not need to complete knowledge graphs manually. A great amount of manpower can be saved, and work efficiency can be increased. After the completion process, knowledge graphs can provide volume and valuable human knowledge in a structured way. They can be applied to many scenarios that require human knowledge. For example, an E-commerce company can use knowledge graphs for customer service and personalized recommendation. Medical workers can use them to make a diagnosis.

One ethical concern when using automatic knowledge graph completion methods is the potential for privacy disclosure. If we use public data on the Internet to construct a knowledge graph and then complete it using the proposed method, personal information that one does not want to make public may be unveiled. Therefore, we advise everyone to be cautious about usage scenarios of automatic knowledge graph completion methods.

## Acknowledgments and Disclosure of Funding

We would like to thank all the anonymous reviewers for their insightful comments. This work was supported in part by NSFC (61822604, 61836006, U19B2026).

## Footnotes

[1]For more details about the 10 queries, please refer to the supplementary material.

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
