[Supplementary Material]

# Duality-Induced Regularizer for Tensor Factorization Based Knowledge Graph Completion

# Supplementary Material

**Zhanqiu Zhang**     **Jianyu Cai**     **Jie Wang** [*]
University of Science and Technology of China
{zzq96,jycai}@mail.ustc.edu.cn,jiewangx@ustc.edu.cn

## 1  Proof for Theorem 1

**Theorem 1.** *Suppose that $\hat{\mathcal{X}}_j = \boldsymbol{H}\boldsymbol{R}_j\boldsymbol{T}^\top$ for $j = 1, 2, \ldots, |\mathcal{R}|$, where $\boldsymbol{H}, \boldsymbol{T}, \boldsymbol{R}_j$ are real matrices and $\boldsymbol{R}_j$ is diagonal. Then, the following equation holds*

$$\min_{\hat{\mathcal{X}}_j = \boldsymbol{H}\boldsymbol{R}_j\boldsymbol{T}^\top} \frac{1}{\sqrt{|\mathcal{R}|}} \sum_{j=1}^{|\mathcal{R}|} (\|\boldsymbol{H}\boldsymbol{R}_j\|_F^2 + \|\boldsymbol{T}\|_F^2 + \|\boldsymbol{T}\boldsymbol{R}_j^\top\|_F^2 + \|\boldsymbol{H}\|_F^2) = \|\hat{\mathcal{X}}\|_*.$$

*The equation holds if and only if $\|\boldsymbol{h}_{:d}\|_2\|\boldsymbol{r}_{:d}\|_2 = \sqrt{|\mathcal{R}|}\|\boldsymbol{t}_{:d}\|_2$ and $\|\boldsymbol{t}_{:d}\|_2\|\boldsymbol{r}_{:d}\|_2 = \sqrt{|\mathcal{R}|}\|\boldsymbol{h}_{:d}\|_2$, for all $d \in \{1, 2, \ldots, D\}$, where $\boldsymbol{h}_{:d}$, $\boldsymbol{r}_{:d}$, and $\boldsymbol{t}_{:d}$ are the $d$-th columns of $\boldsymbol{H}$, $\widetilde{\boldsymbol{R}}$, and $\boldsymbol{T}$, respectively.*

*Proof.* We have that

$$\sum_{j=1}^{|\mathcal{R}|} \left( \|\mathbf{H}\mathbf{R}_j\|_F^2 + \|\mathbf{T}\|_F^2 \right)$$

$$= \sum_{j=1}^{|\mathcal{R}|} \left( \sum_{i=1}^{I} \|\mathbf{h}_i \circ \mathbf{r}_j\|_F^2 + \sum_{d=1}^{D} \|\mathbf{t}_{:d}\|_F^2 \right)$$

$$= \sum_{j=1}^{|\mathcal{R}|} \left( \sum_{d=1}^{D} \|\mathbf{t}_{:d}\|_F^2 + \sum_{i=1}^{I} \sum_{d=1}^{D} \mathbf{h}_{id}^2 \mathbf{r}_{jd}^2 \right)$$

$$= \sum_{j=1}^{|\mathcal{R}|} \sum_{d=1}^{D} \|\mathbf{t}_{:d}\|_2^2 + \sum_{d=1}^{D} \|\mathbf{h}_{:d}\|_2^2 \|\mathbf{r}_{:d}\|_2^2$$

$$= \sum_{d=1}^{D} (\|\mathbf{h}_{:d}\|_2^2 \|\mathbf{r}_{:d}\|_2^2 + |\mathcal{R}| \|\mathbf{t}_{:d}\|_2^2)$$

$$\geq \sum_{d=1}^{D} 2\sqrt{|\mathcal{R}|} \|\mathbf{h}_{:d}\|_2 \|\mathbf{r}_{:d}\|_2 \|\mathbf{t}_{:d}\|_2$$

$$= 2\sqrt{|\mathcal{R}|} \sum_{d=1}^{D} \|\mathbf{h}_{:d}\|_2 \|\mathbf{r}_{:d}\|_2 \|\mathbf{t}_{:d}\|_2.$$

---

[*]Corresponding author.

The equality holds if and only if $\|\mathbf{h}_{:d}\|_2^2\|\mathbf{r}_{:d}\|_2^2 = |\mathcal{R}|\|\mathbf{t}_{:d}\|_2^2$, i.e., $\|\mathbf{h}_{:d}\|_2\|\mathbf{r}_{:d}\|_2 = \sqrt{|\mathcal{R}|}\|\mathbf{t}_{:d}\|_2$.

For all CP decomposition $\hat{\mathcal{X}} = \sum_{d=1}^{D} \mathbf{h}_{:d} \otimes \mathbf{r}_{:d} \otimes \mathbf{t}_{:d}$, we can always let $\mathbf{h}'_{:d} = \mathbf{h}_{:d}$, $\mathbf{r}'_{:d} = \sqrt{\frac{\|\mathbf{t}_d\|_2\sqrt{|\mathcal{R}|}}{\|\mathbf{h}_{:d}\|_2\|\mathbf{r}_{:d}\|_2}}\mathbf{r}_{:d}$ and $\mathbf{t}'_{:d} = \sqrt{\frac{\|\mathbf{h}_{:d}\|_2\|\mathbf{r}_{:d}\|_2}{\|\mathbf{t}_{:d}\|_2\sqrt{|\mathcal{R}|}}}\mathbf{t}_{:d}$ such that

$$\|\mathbf{h}'_{:d}\|_2\|\mathbf{r}'_{:d}\|_2 = \sqrt{|\mathcal{R}|}\|\mathbf{t}'_{:d}\|_2,$$

and meanwhile ensure that $\hat{\mathcal{X}} = \sum_{d=1}^{D} \mathbf{h}'_{:d} \otimes \mathbf{r}'_{:d} \otimes \mathbf{t}'_{:d}$. Therefore, we know that

$$\frac{1}{\sqrt{|\mathcal{R}|}} \sum_{j=1}^{|\mathcal{R}|} \|\hat{\mathcal{X}}_j\|_* = \frac{1}{2\sqrt{|\mathcal{R}|}} \sum_{j=1}^{|\mathcal{R}|} \min_{\hat{\mathcal{X}}_j = \mathbf{HR}_j\mathbf{T}^\top} (\|\mathbf{HR}_j\|_F^2 + \|\mathbf{T}\|_F^2)$$

$$\leq \frac{1}{2\sqrt{|\mathcal{R}|}} \min_{\hat{\mathcal{X}}_j = \mathbf{HR}_j\mathbf{T}^\top} \sum_{j=1}^{|\mathcal{R}|} (\|\mathbf{HR}_j\|_F^2 + \|\mathbf{T}\|_F^2)$$

$$= \min_{\hat{\mathcal{X}} = \sum_{d=1}^{D} \mathbf{h}_{:d} \otimes \mathbf{r}_{:d} \otimes \mathbf{t}_{:d}} \sum_{d=1}^{D} \|\mathbf{h}_{:d}\|_2\|\mathbf{r}_{:d}\|_2\|\mathbf{t}_{:d}\|_2$$

$$= \|\hat{\mathcal{X}}\|_*.$$

In the same manner, we know that

$$\frac{1}{2\sqrt{|\mathcal{R}|}} \min_{\hat{\mathcal{X}}_j = \mathbf{HR}_j\mathbf{T}^\top} \sum_{j=1}^{|\mathcal{R}|} (\|\mathbf{TR}_j^\top\|_F^2 + \|\mathbf{H}\|_F^2) = \|\hat{\mathcal{X}}\|_*.$$

The equality holds if and only if $\|\mathbf{t}_{:d}\|_2\|\mathbf{r}_{:d}\|_2 = \sqrt{|\mathcal{R}|}\|\mathbf{h}_{:d}\|_2$.

Therefore, the conclusion holds if and only if $\|\mathbf{h}_{:d}\|_2\|\mathbf{r}_{:d}\|_2 = \sqrt{|\mathcal{R}|}\|\mathbf{t}_{:d}\|_2$ and $\|\mathbf{t}_{:d}\|_2\|\mathbf{r}_{:d}\|_2 = \sqrt{|\mathcal{R}|}\|\mathbf{h}_{:d}\|_2$, $\forall d \in \{1, 2, \ldots, D\}$. $\qquad\square$

Therefore, for DURA, we know that

$$\min_{\hat{\mathcal{X}}_j = \mathbf{HR}_j\mathbf{T}^\top} \frac{1}{\sqrt{|\mathcal{R}|}} g(\hat{\mathcal{X}}) = \|\hat{\mathcal{X}}\|_*,$$

which completes the proof.

Table 1: Comparison to Reg_p1. "R": RESCAL. "C": ComplEx.

| | WN18RR | | | FB15k-237 | | |
|---|---|---|---|---|---|---|
| | MRR | H@1 | H@10 | MRR | H@1 | H@10 |
| R-Reg_p1 | .281 | .220 | .394 | .310 | .228 | .338 |
| C-Reg_p1 | .409 | .393 | .439 | .316 | .229 | .487 |
| R-DURA | **.498** | **.455** | .577 | .368 | **.276** | .550 |
| C-DURA | .491 | .449 | .571 | **.371** | **.276** | **.560** |

## 2 The optimal value of p

In DB models, the commonly used $p$ is either 1 or 2. When $p = 2$, DURA takes the form as the one in Equation (8) in the main text. If $p = 1$, we cannot expand the squared score function of the associated DB models as in Equation (4). Thus, the induced regularizer takes the form of $\sum_{(h_i, r_j, t_k) \in \mathcal{S}} \|\mathbf{h}_i\bar{\mathbf{R}}_j - \mathbf{t}_k\|_1 + \|\mathbf{t}_k\mathbf{R}_j^\top - \mathbf{h}_i\|_1$. The above regularizer with $p = 1$ (Reg_p1) does not gives an upper bound on the tensor nuclear-2 norm as in Theorem 1. Table 1 shows that, DURA significantly outperforms Reg_p1 on WN18RR and FB15k-237. Therefore, we choose $p = 2$.

Table 2: Hyperparameters found by grid search. $k$ is the embedding size, $b$ is the batch size, $\lambda$ is the regularization coefficients, and $\lambda_1$ and $\lambda_2$ are weights for different parts of the regularizer.

| | WN18RR | | | | | FB15k-237 | | | | | YAGO3-10 | | | | |
|---|---|---|---|---|---|---|---|---|---|---|---|---|---|---|---|
| | $k$ | $b$ | $\lambda$ | $\lambda_1$ | $\lambda_2$ | $k$ | $b$ | $\lambda$ | $\lambda_1$ | $\lambda_2$ | $k$ | $b$ | $\lambda$ | $\lambda_1$ | $\lambda_2$ |
| CP | 2000 | 100 | 1e-1 | 0.5 | 1.5 | 2000 | 100 | 5e-2 | 0.5 | 1.5 | 1000 | 1000 | 5e-3 | 0.5 | 1.5 |
| ComplEx | 2000 | 100 | 1e-1 | 0.5 | 1.5 | 2000 | 100 | 5e-2 | 0.5 | 1.5 | 1000 | 1000 | 5e-2 | 0.5 | 1.5 |
| RESCAL | 512 | 1024 | 1e-1 | 1.0 | 1.0 | 512 | 512 | 1e-1 | 2.0 | 1.5 | 512 | 1024 | 5e-2 | 1.0 | 1.0 |

# 3 Computational Complexity

Suppose that $k$ is the number of triplets known to be true in the knowledge graph, $n$ is the embedding dimension of entities. Then, for CP and ComplEx, the complexity of DURA is $O(kn)$; for RESCAL, the complexity of DURA is $O(kn^2)$. That is to say, the computational complexity of weighted DURA is the same as the weighted squared Frobenius norm regularizer.

# 4 More Details About Experiments

In this section, we introduce the training protocol and the evaluation protocol.

## 4.1 Training Protocol

We adopt the cross entropy loss function for all considered models as suggested in Ruffinelli et al. [5]. We adopt the "reciprocal" setting that creates a new triplet $(e_k, r_j^{-1}, e_i)$ for each triplet $(e_i, r_j, e_k)$ [4, 3]. We use Adagrad [2] as the optimizer, and use grid search to find the best hyperparameters based on the performance on the validation datasets. Specifically, we search learning rates in $\{0.1, 0.01\}$, regularization coefficients in $\{0, 1 \times 10^{-3}, 5 \times 10^{-3}, 1 \times 10^{-2}, 5 \times 10^{-2}, 1 \times 10^{-1}, 5 \times 10^{-1}\}$. On WN18RR and FB15k-237, we search batch sizes in $\{100, 500, 1000\}$ and embedding sizes in $\{500, 1000, 2000\}$. On YAGO3-10, we search batch sizes in $\{256, 512, 1024\}$ and embedding sizes in $\{500, 1000\}$. We search both $\lambda_1$ and $\lambda_2$ in $\{0.5, 1.0, 1.5, 2.0\}$.

We implement DURA in PyTorch and run on all experiments with a single NVIDIA GeForce RTX 2080Ti graphics card.

As we regard the link prediction as a multi-class classification problem and adopt the cross entropy loss, we can assign different weights for different classes (i.e., tail entities) based on their frequency of occurrence in the training dataset. Specifically, suppose that the loss of a given query $(h, r, ?)$ is $\ell((h, r, ?), t)$, where $t$ is the true tail entity, then the weighted loss is

$$w(t)\ell((h, r, ?), t),$$

where

$$w(t) = w_0 \frac{\#t}{\max\{\#t_i : t_i \in \text{training set}\}} + (1 - w_0),$$

$w_0$ is a fixed number, $\#t$ denotes the frequency of occurrence in the training set of the entity $t$. For all models on WN18RR and RESCAL on YAGO3-10, we choose $w_0 = 0.1$ and for all the other cases we choose $w_0 = 0$.

We choose a learning rate of $0.1$ after grid search. Table 2 shows the other best hyperparameters for DURA found by grid search. Please refer to the Experiments part in the main text for the search range of the hyperparameters.

## 4.2 Evaluation Protocol

Following Bordes et al. [1], we use entity ranking as the evaluation task. For each triplet $(h_i, r_j, t_k)$ in the test dataset, the model is asked to answer $(h_i, r_j, ?)$ and $(t_k, r_j^{-1}, ?)$. To do this, we fill the

positions of missing entities with candidate entities to create a set of candidate triplets, and then rank the triplets in descending order by their scores. Following the "Filtered" setting in Bordes et al. [1], we then filter out all existing triplets known to be true at ranking. We choose Mean Reciprocal Rank (MRR) and Hits at N (H@N) as the evaluation metrics. Higher MRR or H@N indicates better performance. Detailed definitions are as follows.

- The mean reciprocal rank is the average of the reciprocal ranks of results for a sample of queries Q:

$$\text{MRR} = \frac{1}{|Q|} \sum_{i=1}^{|Q|} \frac{1}{\text{rank}_i}.$$

- The Hits@N is the ratio of ranks that no more than $N$:

$$\text{Hits@N} = \frac{1}{|Q|} \sum_{i=1}^{|Q|} \mathbb{1}_{x \leq N}(\text{rank}_i),$$

where $\mathbb{1}_{x \leq N}(\text{rank}_i) = 1$ if $\text{rank}_i \leq N$ or otherwise $\mathbb{1}_{x \leq N}(\text{rank}_i) = 0$.

### 4.3 The queries in T-SNE visualization

In Table 3, we list the ten queries used in the T-SNE visualization (Section 5.4 in the main text). Note that a query is represented as $(h, r, ?)$, where $h$ denotes the head entity and $r$ denotes the relation.

Table 3: The queries in T-SNE visualizations.

| Index | Query |
|-------|-------|
| 1 | (political drama, /media_common/netflix_genre/titles, ?) |
| 2 | (Academy Award for Best Original Song, /award/award_category/winners./award/award_honor/ceremony,?) |
| 3 | (Germany, /location/location/contains,?) |
| 4 | (doctoral degree , /education/educational_degree/people_with_this_degree./education/education/major_field_of_study,?) |
| 5 | (broccoli, /food/food/nutrients./food/nutrition_fact/nutrient,?) |
| 6 | (shooting sport, /olympics/olympic_sport/athletes./olympics/olympic_athlete_affiliation/country,?) |
| 7 | (synthpop, /music/genre/artists, ?) |
| 8 | (Italian American, /people/ethnicity/people,?) |
| 9 | (organ, /music/performance_role/track_performances./music/track_contribution/role, ?) |
| 10 | (funk, /music/genre/artists, ?) |