[Reviews · NeurIPS 2020]

Review 1

Summary and Contributions: In this work, the authors propose a new regularizer for tensor factorization-based knowledge graph completion. The new regularizer is motivated by the equivalence (duality) between the new loss function and a Frobenius norm-based distance that needs to be minimized. In this sense, the paper provides very nice motivation and justification for the new method which outperforms other tensor factorization-based approaches as the experimental results demonstrated. The Authors have answered satisfactorily to all my questions and concerns even providing new comparison results. I thank the Authors for such detailed responses.

Strengths: - I found the results significant and relevant to the NeurIPS community - Very well written. The problem is nicely introduced, and the new method is well motivated through the duality of the loss and a Frobenius norm-base distance.

Weaknesses: - It is still not clear how far are the obtained results compared to state of the art distance-based methods. All experiments are performed only with tensor-based methods using different regularizers. - Some improvements can be incorporated (see detailed comments below)

Correctness: Yes, however it could be improved (see detailed comments below)

Clarity: Yes, the problem is nicely introduced and the new solution is well justified.

Relation to Prior Work: Yes

Reproducibility: Yes

Additional Feedback: - Line 70: DB methods are based on Minkowski distance, however, in this paper the duality is stablished only for the case of Frobenius norm, i.e. Minkowski distance with p=2. It would be nice that authors provide a deeper explanation about the role on parameter p in DB methods. What is the optimal value of p in state of the arts methods? - Line 139: I would suggest changing “When we train a distance based model” to “When we train an Euclidean distance based model” - Line 180: I think there is an error. The sentence: “the regularizer 5 and 6” should be changed to “the regularizer 4 and 5” (check equation numbering) - Line 180: As a regularizer having several terms, it would be convenient to consider different regularizer coefficients as hyperparameters. In fact, in supplemental material (lines 36-37) the cost function has 3 hyperparameters: lambda, lambda_1 and lambda_2. Please, clarify this in the main document and provide some guidance on how to chose hyperparameters. - In Section 4.4, it is demonstrated that when relation matrices are diagonal, the proposed new regularizer is equivalent to the nuclear-2 norm. What is the effect of having non-diagonal relation matrices? Is it still favoring minimum nuclear-2 norm solutions? I think it would be useful to analyze it at least numerically. - In Experiments, why there is no comparison against a distance-based method? I wondering if methods considered in the experiments can be considered as state of the art results. - Please, provide definition of used performance measures (MRR and H@k)


Review 2

Summary and Contributions: The paper presents a novel regularization approach for tensor factorization based knowledge graph completion (KGC) models. There are two key contributions presented in this work: 1. The paper established a duality relationship between the tensor factorization-based KGC model (primal) and the distance-based KGC model (dual). 2. Based on duality together with tensor factorization, the paper presents a DUality-induced RegulArizer (DURA), which can be widely applicable to tensor factorization based knowledge graph completion models.

Strengths: 1. Nice problem, and elegant approach. 2. Mostly thorough evaluation. 3. Algorithm seems to perform well.

Weaknesses: 1. Writing was confusing in some places; further explanation was needed. 2. More recent comparison methods for knowledge graph completion could be included.

Correctness: Technically sound and correct.

Clarity: Generally well written with some minor blemishes.

Relation to Prior Work: The previous work is appropriately cited and discussed.

Reproducibility: Yes

Additional Feedback: 1. The writing was a bit confusing at times. In particular, there should be a bit more explanation of why the number of entities in the WN18RR dataset is smaller than that reported in the original paper [22]; in page 10, line 193, it is unclear what is being said here of A(i, j), but A seems never used in the paper. 2. There should be more recent KGC algorithms included in the evaluation. The experiments were mostly focused on self-assessment to compare - with or without DURA, or other simple regularizers. There have been dozens of very good KGC algorithms presented recently. In particular, the propsoed method focused on two-way coupling effects, while the paper in [1] directly used a Tucker-based KGC model to explore three-way interactions, it would be better to make a comparison of them. [1] I. Balazevic, C. Allen, and T. M. Hospedales, “TuckER: Tensor factorization for knowledge graph completion,” in EMNLPIJCNLP, 2019, pp. 5185–5194. Minor: In page 8, line 180, "the regularizer 5 and 6" should be "the regularizers (4) and (5)". I think the authors provided an articulated answer to our reviews. The new experimental results are convincing and the analysis of these results is insightful and helpful for other researchers.


Review 3

Summary and Contributions: In this paper, authors propose duality-induced regulariser (DURA), a way of regularising tensor factorisation-based neural link predictors. The underlying idea is that, if multiple entities appear as objects in a (s, p, ?) triple, they should be close in embedding space, and authors enforce this behaviour by regularising the interactions between the latent factors.

Strengths: Interesting ideas and clear write-up, although I think it is imprecise in some places - see my comments in the "Correctness" section.

Weaknesses: Results do not seem significantly better than these reported in e.g. https://arxiv.org/abs/1806.07297 (https://github.com/facebookresearch/kbc/)

Correctness: One issue I have with this paper is that it broadly refers tensor factorisation-based models while it actually talks about ComplEx, e.g. see lines 130-131. The derivation of DURA is not totally clear to me: if you want your model to behave like in the Eq. on line 134, why just not use that scoring function instead? Authors state, on lines 160-161: "Therefore, the performance of TFB models for KBC is usually unsatisfying", but it doesn't look like that to me, since their methods is on par with simpler TFB models such as https://arxiv.org/abs/1806.07297 (see the result tables in https://github.com/facebookresearch/kbc/) Lines 254-255: "Note that [..] RESCAL, we only compare DURA tot he squared Frobenius norm regularisation as N3 does not apply to it." - why? You can use the N3 norm on RESCAL's relation embedding matrix as well.

Clarity: Apart from some imprecisions, the paper is very readable

Relation to Prior Work: There is plenty of work on regularising factorisation models and KG embeddings, this could probably be expanded quite a bit. For instance, in https://arxiv.org/abs/1707.07596 authors use background knowledge for regularising knowledge graph embedding models.

Reproducibility: Yes

Additional Feedback:


Review 4

Summary and Contributions: This paper proposes a regularization scheme for tensor-decomposition based KG completion methods. The regularization scheme is well-motivated, albeit very similar to L2 regularization, and the paper gives good intuitions for why the regularization might be helpful

Strengths: Here's some of the things I liked about this paper: 1. I liked section 4.2 that pictorially shows how DURA can encourage embeddings that occur in similar (subject, relation) contexts to be close together. 2. The analysis that shows that DURA regularized embeddings are more robust to sparsity

Weaknesses: I think a lot of comparisons / qualitative analysis is missing: 1. I would like to see a T-SNE plot that shows that the claim that entities with similar (e1,r) contexts are indeed being assigned similar representations is true. 2. There have been a few new KG completion models in the past few years such as KBAT / GAATs that achieve competitive performance on WN18RR/FB15k-237. Would DURA help on these methods as well? I think it's ok if the answer is no, but it would make the paper stronger if a discussion on limitations is added.

Correctness: Yes

Clarity: Yes

Relation to Prior Work: Some of the recent work on KG completion has been omitted. Some examples: 1.Learning Attention-based Embeddings for Relation Prediction in Knowledge Graphs. Nathani et al. ACL 2019 2. Quaternion Knowledge Graph Embeddings. Zhang et al. NeurIPS 2019

Reproducibility: Yes

Additional Feedback:

[Author Response · NeurIPS 2020]

We thank the reviewers for their insightful comments.

**Response to Reviewer 1**

**Comparison with SOTA distance-based methods** We compare
our models with some SOTA distance-based (DB) models, includ-
ing RotatE [21], MuRP [Ref1], and HAKE [30]. Table 1 shows
that, RESCAL-DURA and ComplEx-DURA perform competitively
with the SOTA DB models. RESCAL-DURA outperforms all the
aforementioned DB models in terms of MRR and H@1.

**The optimal value of p** In DS models, the commonly used $p$ is either
1 or 2. When $p = 2$, DURA takes the form as the one in line 180. If
$p = 1$, we cannot expand the squared score function of the associated
DS models as in line 138. Thus, the induced regularizer takes the form of $\sum_{(h_i, r_j, t_k) \in \mathcal{S}} \|\mathbf{h}_i \bar{\mathbf{R}}_j - \mathbf{t}_k\|_1 + \|\mathbf{t}_k \mathbf{R}_j^\top - \mathbf{h}_i\|_1$.
The above regularizer with $p = 1$ (Reg_p1) does not gives an upper bound on the tensor nuclear-2 norm as in Theorem
1. Moreover, experiments show that, DURA significantly outperforms Reg_p1 on WN18RR and FB15k-237 (see the
third and fourth parts of Table 1). Therefore, we choose $p = 2$.

**Analyses for non-diagonal relation matrices** Tensor nuclear-2 norm is defined based on the CP decomposition [6].
When relation matrices are non-diagonal, TFB models do not take the form of the CP decomposition. Therefore,
Theorem 1 does not apply to the non-diagonal case. Moreover, as stated in [6], computing the nuclear-2 norm of a
3-tensor over $\mathbb{R}$ is NP-hard, which implies that numerical analyses are intractable.

**Other suggestions** We will improve our paper accordingly.

**Response to Reviewer 2**

**Confusing writing at times** WN18RR used in our paper is the same as that in the original paper [22], of which the
number of entities is $40,943$. Thanks for pointing out this typo. We will correct it accordingly. In line 193, $A$ is a
placeholder that can be any matrix. That is, for any matrix $A$, $A(i, j)$ represents the entry in the $i$-th row and the $j$-th
column of it. We will polish it in the final submission, if accepted.

**More recent KGC algorithms included in the evaluation** Table 1 shows the evaluation results of our methods
against recent DB models and TuckER [Ref2]. We will include the results in the final submission, if accepted.

**Response to Reviewer 3**

**Tensor factorization-based (TFB) models in this paper is actually ComplEx** TFB models in our paper (e.g., lines
130-131) can be CP, ComplEx, and RESCAL. Note that, both the real part and the conjugate of a real matrix are equal to
the matrix itself. When all the embeddings are real, the score function in lines 130-131 corresponds to CP or RESCAL.

**Why just not use the score function in line 134 instead** As the regularization coefficient is usually small, a TFB
model regularized by DURA is not the same as its associated DB model. The regularized model is dominant by the
score function of the TFB model. Therefore, DURA does not aim to make models behave like the score function in
line 134. Instead, it aims at introducing the prior knowledge that, tail (head) entities—connected to a head (tail) entity
through the same relation—should have similar embeddings. Hence, we cannot just use the score function in line 134.

**The statements in lines 160-161 are imprecise** "TFB models" in these lines corresponds to TFB models without any
regularization. Thus, it does not include models with N3. We will provide more details in the final version, if accepted.

**RESCAL+N3** The definition of N3 regularization [13] depends on a summation of $R$ vector norms, where $R$ is the
tensor rank (see Section 4.1 in [13]). Note that, tensor ranks used in [13] are defined based on the CP decomposition.
As RESCAL does not take the form of the CP decomposition, we cannot apply N3 regularization to RESCAL.

**Prior work on regularizing factorization models and KGE** We will cite these papers in the final version, if accepted.

**Response to Reviewer 4**

**T-SNE plot** Suppose that $(h_i, r_j)$ is a *query*, where $h_i$ and $r_j$ are head entities
and relations, respectively. An entity $t_k$ is an *answer* to a query $(h_i, r_j)$ if
$(h_i, r_j, t_k)$ is valid. We randomly selected 10 queries in FB15k-237, each of
which has more than 50 answers. Then, we use T-SNE to visualize the answers'
embeddings generated by CP and CP-DURA. Figure 1 shows that, with DURA,
entities with the same $(h_i, r_j)$ contexts are indeed being assigned more similar
representations. We will include the results in the final version, if accepted.

**Would DURA help on other KGC models such as KBAT/GAATs** The an-
swer is no. DURA is designed for tensor factorization based models and can bring significant improvements for them.
However, it does not apply to models in other categories, such as KBAT/GAATs. We will discuss the potential limitation
of DURA in the final submission, if this paper is accepted.

**Some of the recent work on KGC has been omitted** We will cite all these papers in the final submission, if accepted.

[Ref1] I. Balaževiá, C. Allen, and T. M. Hospedales. Multi-relational Poincaré Graph Embeddings. NeurIPS 2019.

[Ref2] I. Balaževiá, C. Allen, and T. M. Hospedales. TuckER: Tensor factorization for knowledge graph completion. EMNLP 2019.

Table 1: Comparison to other SOTA models. "R": RESCAL. "C": ComplEx.

|  | WN18RR | | | FB15k-237 | | |
|---|---|---|---|---|---|---|
|  | MRR | H@1 | H@10 | MRR | H@1 | H@10 |
| RotatE | .476 | .428 | .571 | .338 | .241 | .533 |
| MuRP | .481 | .440 | .566 | .335 | .243 | .518 |
| HAKE | .497 | .452 | **.582** | .346 | .250 | .542 |
| TuckER | .470 | .443 | .526 | .358 | .266 | .544 |
| R-Reg_p1 | .281 | .220 | .394 | .310 | .228 | .338 |
| C-Reg_p1 | .409 | .393 | .439 | .316 | .229 | .487 |
| R-DURA | **.498** | **.455** | .577 | .368 | **.276** | .550 |
| C-DURA | .491 | .449 | .571 | **.371** | **.276** | **.560** |

(a) CP      (b) CP-DURA

Figure 1: Plots of tail entity embeddings.

[Meta-Review · NeurIPS 2020]

There are roughly two different approaches in the literature for knowledge graph completion (KGC), namely distance based (DB) models and tensor factorization based (TFB) models. Although both approaches have their own advantages and disadvantages over each other, TFB models cannot attain state-of-the-art performance due to overfitting problem, and therefore various regularizers are employed for TFB models. In the paper, authors propose a regularizer for TFB models, namely Duality-induced Regularization (DURA), which is inspired by the score functions of the DB models. They come up with a dual problem which involves a distance based KGC model, and show that when the aforementioned regularizer is employed for the primal problem (i.e. TFB model), both problems become equivalent. By doing so, they are able to shed light on the connections between TFB and DB models, as well as attaining a much better performance on TFB models. The rebuttal has been found quite useful by the reviewers to resolve issues. The experimental results are also found convincing. The paper has several strong points - Well written and the main idea of the paper is easy to grasp and all the related technical concepts are explained in a clear way. - Empirical evaluation shows that the proposed regularization methodology significantly increases the performances of the tensor factorization based models. When compared to some other regularization methods, DURA performs still better in general and it finds sparser embeddings for the entities. - Proposed regularization for tensor factorization based models is found in a very intuitive way, i.e. by introducing a distance based dual model. Thereby, it establishes an implicit connection between tensor based models and distance based models. - The implications of why employing DURA may help to prevent overfitting is discussed in great detail. - When the relation embedding matrices are diagonal, DURA gives an upper bound tensor nuclear 2-norm of the observed binary tensor, which in turn provides further implications about how DURA may help tensor factorization based knowledge graph completion. A number of weaknesses have also been mentioned but the authors have successfully answered many of these.